# Identification and Analysis of Sex-Biased MicroRNAs in Human Diseases

**DOI:** 10.3390/genes14091688

**Published:** 2023-08-25

**Authors:** Bitao Zhong, Chunmei Cui, Qinghua Cui

**Affiliations:** Department of Biomedical Informatics, Center for Noncoding RNA Medicine, State Key Laboratory of Vascular Homeostasis and Remodeling, School of Basic Medical Sciences, Peking University, 38 Xueyuan Rd, Beijing 100191, China; zhongbt@pku.edu.cn (B.Z.); ccm328@bjmu.edu.cn (C.C.)

**Keywords:** microRNA, sex-biased expression, bioinformatic analysis

## Abstract

It is well known that significant differences exist between males and females in both physiology and disease. Thus, it is important to identify and analyze sex-biased miRNAs. However, previous studies investigating sex differences in miRNA expression have predominantly focused on healthy individuals or restricted their analysis to a single disease. Therefore, it is necessary to comprehensively identify and analyze the sex-biased miRNAs in diseases. For this purpose, in this study, we first identified the miRNAs showing sex-biased expression between males and females in diseases based on a number of miRNA expression datasets. Then, we performed a bioinformatics analysis for these sex-biased miRNAs. Notably, our findings revealed that women exhibit a greater number of conserved miRNAs that are highly expressed compared to men, and these miRNAs are implicated in a broader spectrum of diseases. Additionally, we explored the enriched transcription factors, functions, and diseases associated with these sex-biased miRNAs using the miRNA set enrichment analysis tool TAM 2.0. The insights gained from this study could carry implications for endeavors such as precision medicine and possibly pave the way for more targeted and tailored approaches to disease management.

## 1. Introduction

The differences between men and women are evident in both physiological and pathological processes, emphasizing the importance of precision medicine in developing more accurate and tailored treatments for each sex [1]. Therefore, it is quite important to dissect the sex-biased molecules in physiology and disease and systematically analyze them. Previous studies have found many variations in disease development and treatment responses between men and women, underscoring the need to identify and analyze these differences to optimize modern medical strategies [2,3]. MicroRNAs (miRNAs) are one class of short noncoding RNA molecules that do not encode proteins but perform crucial functions in regulating various cellular activities and physiological processes at the post-transcriptional level, including cell growth, differentiation, apoptosis, immune and inflammation response, development, and metabolism [4]. Therefore, miRNA-related dysfunction often involves various human diseases, including cancer, cardiovascular disease, neurodegenerative disease, etc. Like protein-coding genes, miRNAs could also show sex-bias in physiology and human disease. Indeed, previous studies have suggested that miRNA expression levels differ between males and females across various tissues [5], and these sex-specific miRNA expressions may have important functional implications.

For example, a recent study by Deny et al. found that miRNA expression profiles differed between males and females during cystic fibrosis [6], while Gunay et al. found significant miRNA differences between mice of different sexes in the methanol intoxication state [7]. Kahnamoui et al. found that respiratory sensitivity was associated with miRNA expression levels in mice of different sexes [8]; Rosenberg et al. found an association between the effects of different sexes on smoking and lung development [9]. In addition, research showed that certain miRNAs are expressed differently in different sexes during fetal development [10,11], which, in turn, results in different developmental processes. There have even been studies of male–female gender differences in miRNA-associated methylation modifications during solid tumor pathology [12]. In addition, countless studies have analyzed sex-differentiated miRNAs for a single type of disease or have used the expression profiles of miRNAs from non-human sources [13,14,15,16].

However, these studies primarily focused on investigating sex differences in miRNA expression among healthy subjects, and the investigation into human sex-specific miRNA expression patterns in pathological states has largely been overlooked. To better study the major classes of diseases, we should comprehensively collect miRNA expression datasets in human disease, which include gender information and identify, and analyze the sex-biased miRNAs in human disease. To this end, here, we collected a series of miRNA expression datasets containing both disease and gender information from the GEO database. Then, we identified the miRNAs, which are differentially expressed between males and females in specific diseases. These miRNAs are the disease-associated sex-biased miRNAs. Subsequently, bioinformatics analyses were conducted to unveil these disease-associated sex-biased miRNA expression patterns in various pathological states.

## 2. Materials and Methods

### 2.1. Classification Criteria of the Disease

We classified these collected datasets using the International Classification of Diseases standard ICD-11 published by WHO (https://icd.who.int/en (accessed on 24 March 2023)). The collected diseases were classified into 8 major categories: Infectious (GSE34608, GSE134358), Neoplasms (GSE42657, GSE25631, GSE142699, GSE39040, GSE39052, GSE124678, GSE126093, GSE68204, GSE75283, GSE76903, GSE137140, GSE102286, GSE155209, GSE145259), Endocrine diseases (GSE41223), Mental disorders (GSE167559), Epilepsy (GSE199759, GSE205661), Retinopathy (GSE160308), Developmental anomalies (GSE137995, GSE215940), and Injury (GSE174661), and we scrutinized 4 of the categories that had more significant differences (Infectious, Neoplasms, Endocrine diseases, Developmental anomalies).

### 2.2. Appraisal of the Disease-Associated Sex-Biased miRNAs

The GEO (Gene Expression Omnibus, https://www.ncbi.nlm.nih.gov/geo/ (accessed on 9 February 2023)) database, a large gene sequencing database and a public functional genomics data repository provided by NCBI (National Center for Biotechnology Information), contains a wealth of sequencing results. We conducted a comprehensive search for miRNA expression datasets in the GEO database that contained pertinent information on both sex and human disease samples. We only used the normalized miRNA expression profiles for each dataset. We used the Wilcoxon Rank Sum test to identify these sex-biased miRNAs, dismissing fold changes to retain sufficient miRNA for analysis. As a result, our analysis identified sex-biased miRNAs in four classes of samples: peripheral blood, brain, neoplasms tissue, and other tissues. Following that, we manually curated the datasets that met these specific criteria. Upon mapping the probes to miRNA names and filtering out null values, we classified the samples in each dataset into distinct cohorts: female disease and male disease. This systematic grouping allowed us to pinpoint miRNAs that exhibited a gender bias and were associated with specific diseases.

By applying the Wilcoxon test and setting *p* < 0.05 as the threshold value, we identified the differentially expressed miRNAs between males and females as the sex-biased miRNAs. By amalgamating the results from each dataset and excluding miRNAs with conflicting biases, we derived an ultimate list of disease-associated sex-biased miRNAs.

### 2.3. Analysis of Evolutionary Conservation of the Disease-Associated Sex-Biased miRNAs

The evolutionary conservation of miRNAs is an important topic and feature that is highly related to human diseases. As previously noted [5], we classified all of the miRNA genes into five groups of evolutionary conservation: human-specific (G1), primate-specific (G2), mammal-specific (G3), vertebrate-specific (G4), and miRNA genes present in other distal species (G5, the most conserved). In addition to conservation group, the number of species included in the miRNA family may also represent the evolutionary conservation of miRNAs. To calculate the number of species in family members for one specific miRNA, we first downloaded the miRNA family file from the miRBase database (https://www.mirbase.org (accessed on 28 March 2023)) and then calculated the number of species in family members for each human miRNA.

### 2.4. Analysis of TFs and the Function of the Disease-Associated Sex-Biased miRNAs

Like protein-coding genes, miRNA genes are also transcriptionally regulated via transcription factors (TFs). miRNAs and their TFs, thus, form regulatory circuits to play important roles in disease formation and development. Therefore, it is important to analyze the TFs regulating these disease-associated sex-biased miRNAs. Moreover, it is also important to analyze the functions of these disease-associated sex-biased miRNAs. Here, we conducted comprehensive functional enrichment, disease enrichment, and transcription factor enrichment analyses for FBmiR and MBmiR with the help of the process of TAM 2.0 (http://www.lirmed.com/tam2/ (accessed on 29 March 2023)). This tool enabled us to explore and identify enriched functional annotations, disease associations, and transcription factor regulatory networks specifically associated with FBmiR and MBmiR.

## 3. Results

### 3.1. Subsection

#### 3.1.1. Identification of the Disease-Associated Sex-Biased miRNAs

To identify the disease-associated sex-biased miRNAs, i.e., the miRNAs showing differential expression between males and females in specific human diseases, we searched the GEO database for miRNA expression datasets with annotation information, including disease and gender. As a result, we collected 24 datasets (Table 1) for subsequent analysis.

We next identified miRNAs that show significant differences between females and males in the disease state. We named miRNAs with higher expression levels in females in the disease female-biased miRNAs (FBmiR) and those with higher expression levels in males as male-biased miRNAs (MBmiR) in the specific disease. We used the Wilcoxon Rank Sum test to identify these sex-biased miRNAs, using a *p*-value of 0.05 as the threshold and not considering fold changes to retain sufficient miRNA for further analysis. As a result, our analysis identified sex-biased miRNAs in four classes of samples: peripheral blood, brain, tumor tissue, and other tissues. As a result, we identified 1264 FBmiR and 431 MBmiR, respectively. Among them, the groups with a relatively high proportion of FBmiR and MBmiR were the infectious group (21.0%), the neoplasms group (60.0%), the endocrine diseases group (2.3%), and the developmental anomalies group (5.7%). So, further bioinformatics analysis will mainly focus on these groups.

Previous studies have demonstrated that even in a healthy state, the human body exhibits differential expression of certain miRNAs based on gender [17,18]. To maintain methodological consistency, we leveraged gender-specific miRNA expression data from our laboratory’s prior research to establish a comparative framework with miRNAs from the current disease state [5]. By aligning the newly identified miRNAs with their precursor levels, we performed a comparative analysis. Interestingly, the results revealed that only 6 out of 1006 premature miRNAs (0.60%) with high expression overlapped between healthy and diseased states in females. Similarly, in males, the overlap was observed in just 2 out of 387 premature miRNAs (0.52%). These findings underscore the substantial divergence in miRNA expression profiles between healthy and diseased states in the human body. Importantly, the newly discovered disease-associated miRNAs exhibit a significant disparity from those present in healthy states, suggesting their potential relevance to pathological processes.

#### 3.1.2. Analysis of the Chromosomal Arrangement of the Disease-Associated Sex-Biased miRNA Genes

It is well known that genes, including miRNAs, are not randomly located in the human genome and chromosomes. Therefore, it is necessary to investigate the chromosome distribution of the FBmiR and MBmiR. Then, having identified these genes, we first examined the chromosome distribution of these disease-associated sex-biased miRNA genes. Previous studies have shown that miRNAs are located on individual chromosomes [17] and that each chromosome may have a distinct role in pathophysiological processes [18]. Therefore, it is plausible that the distribution of disease-associated sex-biased miRNAs on chromosomes would also vary. To explore this, we compared the distribution of FBmiR and MBmiR on chromosomes (Figure 1a–d). Our findings indicate that in infectious diseases, MBmiR was more abundant than FBmiR in 15 out of the 23 chromosomes, while in neoplasms, FBmiR was more abundant than MBmiR in all 23 chromosomes. This observation suggests that the pathological processes of infection and neoplasms may be oppositely associated in the two sexes. However, it is important to note that the number of MBmiR is higher than that of FBmiR in infectious diseases, whereas in neoplasms, the number of FBmiR is higher than that of MBmiR. This discrepancy may partially explain the aforementioned phenomenon. Moreover, this result suggests that infection and neoplasms may have a deep connection with each other. Furthermore, we observed that the sex-biased miRNAs in infectious diseases were more abundant in chromosomes 8 and 14 than in other chromosomes. In addition, we explored the overlap of these miRNAs in different groups (Figure 1e,f). As much as 87.67% (1258 of 1435) of FBmiR and MBmiR did not overlap in the groups, which shows that it is, therefore, more important for us to analyze the groups of diseases separately.

Additionally, it was reported that the spacing of genes on chromosomes has a significant effect on their function [19]. We then analyzed the intrachromosomal distance between each group of miRNA genes on each specific chromosome. Our results showed that for the infection group, both FBmiR and MBmiR were significantly closer in distance compared to the group of other miRNAs (OmiR) (Figure 2a, FBmiR vs. OmiR, Wilcoxon’s test *p*-value = 2.9 × 10^−5^; MBmiR vs. OmiR, Wilcoxon’s test *p*-value = 9.4 × 10^−9^), while the difference was less significant for the neoplasms group (Figure 2b, FBmiR vs. OmiR, Wilcoxon’s test *p*-value = 0.47; MBmiR vs. OmiR, Wilcoxon’s test *p*-value = 0.023). In addition, FBmiR in endocrine diseases showed a significant difference from OmiR (Figure 2c, FBmiR vs. OmiR, Wilcoxon’s test *p*-value < 2.22 × 10^−15^; MBmiR vs. OmiR, Wilcoxon’s test *p*-value = 0.63), while developmental anomalies did not show a significant difference (Figure 2d, FBmiR vs. OmiR, Wilcoxon’s test *p*-value = 0.46; MBmiR vs. OmiR, Wilcoxon’s test *p*-value = 0.9). This result further suggests that during infection, in both sexes, the expression of certain regions of miRNAs may be more concentrated. This is less pronounced during neoplasm development with weaker immune responses. Previous studies have also shown that the immune response is stronger in women than in men during the process of tumorigenesis, which may be related to the differences in the degree of genetic miRNA aggregation [20]. Therefore, it is likely that disease-associated sex-biased miRNAs could be associated with immune responses.

Lastly, to rule out any potential influence of sex chromosomes on our findings, we analyzed the proportion of disease-associated sex-biased miRNA genes located on autosomes versus sex chromosomes. As a result, we observed that the majority of these miRNA genes were distributed on autosomes but not sex chromosomes (about 93% on autosomes). Together, these results suggest that the distribution of disease-associated sex-biased miRNA genes on chromosomes is not random but appears to be selectively aggregated to serve specific functions.

#### 3.1.3. Analysis of the Evolutionary Conservation of the Disease-Associated Sex-Biased miRNA Genes

Given that evolutionary conservation is an important feature for dissecting the functions and associated diseases of genes, we next consider investigating the evolutionary conservation of these disease-associated sex-biased miRNAs genes. It is known that conserved genes, which are present in relatively old species, often act as essential indicators of gene functionality [21]. It was reported that highly conserved genes tend to encode more basic physiological activities [22]. We then analyzed the percentage distribution of FBmiR, MBmiR, and OmiR in these different miRNA groups (Appendix A). The results show that FBmiR is more prevalent in the more conserved groups for both the infection and neoplasm groups. In contrast, in fast-evolving groups, the proportion of MBmiR is lower, and the proportion of OmiR is higher.

In addition to the conservation group, the number of species included in the miRNA family may also reflect this pattern (Figure 3). The difference was not significant in the infection group diseases (Figure 3a, FBmiR vs. OmiR, Wilcoxon’s test *p*-value = 0.078; MBmiR vs. OmiR, Wilcoxon’s test *p* = 0.38), while the number of species included in the families of both FBmiR and MBmiR in the neoplasm group diseases was significantly higher than that of OmiR (Figure 3b, FBmiR vs. OmiR, Wilcoxon’s test *p* = 1.7 × 10^−5^; MBmiR vs. OmiR, Wilcoxon’s test *p* = 1.5 × 10^−4^). The number of species involved in MBmiR was significantly higher than OmiR in endocrine diseases (Figure 3c, FBmiR vs. OmiR, Wilcoxon’s test *p* = 0.42; MBmiR vs. OmiR, Wilcoxon’s test *p* = 0.0078), while no significant differences were observed in developmental anomalies (Figure 3d, FBmiR vs. OmiR, Wilcoxon’s test *p* = 0.95; MBmiR vs. OmiR, Wilcoxon’s test *p* = 0.54). These results suggest that there may be greater differences between the sexes in basic biological processes during pathological states.

#### 3.1.4. Analysis of Tissue Specificity of the Disease-Associated Sex-Biased miRNAs

The tissue specificity of a gene is closely related to its function and associated diseases, with some genes expressed with tissue specificity. It is then also necessary to investigate whether the conserved nature of these disease-associated sex-biased miRNA genes had some correlation with their tissue specificity. A previous study [23] identified the miRNA expression profiles of different tissues and calculated the tissue specificity index of miRNAs. Accordingly, we obtained the tissue specificity of FBmiR and MBmiR and found that FBmiR showed substantially elevated tissue specificity in contrast to OmiR and MBmiR in the infection group (Figure 4a, FBmiR vs. OmiR, Wilcoxon’s test *p* = 3.1 × 10^−7^; FBmiR vs. MBmiR, Wilcoxon’s test *p* = 8.1 × 10^−8^), while in neoplasm samples, both FBmiR and MBmiR showed higher tissue specificity scores than OmiR (Figure 4b, FBmiR vs. OmiR Wilcoxon’s test *p* = 5.6 × 10^−15^; MBmiR vs. OmiR, Wilcoxon’s test *p* = 0.013) and so in endocrine diseases (Figure 4c, FBmiR vs. OmiR, Wilcoxon’s test *p* = 9.7 × 10^−5^; MBmiR vs. OmiR, Wilcoxon’s test *p* = 0.022). However, no significant differences were observed in the development group (Figure 4d, FBmiR vs. OmiR Wilcoxon’s test *p* = 0.24; MBmiR vs. OmiR, Wilcoxon’s test *p* = 0.78). This suggests that specific tissues of female patients may undergo more intense immune responses when immune reactions occur. Previous studies have suggested that since miRNAs are negatively regulated fragments [24], their tissue specificity may be positively correlated with their conservativeness, which we also concluded computationally.

#### 3.1.5. Analysis of Transcription Factors of the Disease-Associated Sex-Biased miRNAs

As addressed above, miRNAs are transcriptionally regulated via TFs. It is, thus, also important to investigate the TFs of these disease-associated sex-biased miRNAs. So, we next set out to analyze the transcription factors of these disease-associated sex-biased miRNA genes. Using the TAM2.0 tool, we identified the TFs that frequently regulate disease-associated sex-biased miRNAs (Appendix A). As a result, we observed a significant enrichment of TGFB1 in both the immune response group and the neoplasms group within the FBmiR miRNA list. TGFB1 belongs to the TGF-β family and serves as one of the most active multifunctional cellular activity regulators [25]. While TGFB1 acts as a potent growth suppressor in multicellular organisms and significantly inhibits lymphocytes, it also promotes fibroblast cell proliferation, the expression of extracellular matrix proteins, and the synthesis of bone matrix proteins [26]. Therefore, it suggests that the physiological activity of cell proliferation is attenuated in women compared to men in the disease state. Additionally, we identified the enrichment of TP53 and IL6 in the FBmiR miRNA list of the neoplasm group, both of which have been extensively associated with neoplasms in various experiments and are related to tumor immunity [27,28]. This suggests that males and females would exhibit distinct physiological manifestations of pathological processes in tumor immunity.

In addition, the enrichment analysis results are not significant for MBmiR in the neoplasms group or FBmiR and MBmiR in the endocrine diseases and developmental anomalies groups.

#### 3.1.6. Analysis of the Disease Spectrum Width of Disease-Associated Sex-Biased miRNAs

In order to explore the number of diseases associated with these disease-associated sex-biased miRNA genes, we then utilized the disease spectrum width (DSW) score [5] to analyze the association between disease-associated sex-biased miRNAs and different diseases (Figure 5). Notably, OmiR, MBmiR, and FBmiR displayed progressively higher DSW scores, indicating a greater number of associated diseases (Figure 5a, FBmiR vs. OmiR Wilcoxon’s test *p*-value = 4.5 × 10^−8^; MBmiR vs. OmiR Wilcoxon’s test *p*-value = 0.0099; Figure 5b, FBmiR vs. OmiR Wilcoxon’s test *p*-value = 4.4 × 10^−8^; MBmiR vs. OmiR Wilcoxon’s test *p*-value = 0.0082; Figure 5c, FBmiR vs. OmiR Wilcoxon’s test *p*-value = 5.5 × 10^−7^; MBmiR vs. OmiR Wilcoxon’s test *p*-value = 8.1 × 10^−5^; Figure 5d, FBmiR vs. OmiR Wilcoxon’s test *p*-value = 0.048, MBmiR Wilcoxon’s test *p* = 0.88). Higher DSW scores normally suggest a more fundamental function and conservation across species of these genes.

#### 3.1.7. Analysis of the Function of the Disease-Associated Sex-Biased miRNAs

After performing the above analysis, we began to analyze the function of these disease-associated sex-biased miRNAs genes. In order to further explore the potential mechanisms of these disease-associated sex-biased miRNAs, we performed functional enrichment analysis on these miRNAs using the weighted TAM2.0 method (Figure 6). As a result, we observed distinct functional patterns for FBmiR in different groups. In the infection group, FBmiR primarily participates in the apoptosis, immunity, and inflammatory response functions. For example, mir-101-1 is involved in all of the above three processes. Conversely, in the neoplasms group, FBmiR is predominantly associated with cell stemness, cell growth and differentiation, and cell cycle functions. These findings suggest that the disparities in physiological processes related to infection immunity and tumor immunity may be more pronounced in women compared to men, and some of the available studies [29] side step our predictions.

## 4. Discussion

Gender differences permeate various aspects of physiological and pathological processes, making it crucial to discern the physiological and pathological disparities between genders in disease states, especially for new medical technologies. In our previous investigations, we explored sex-biased genes and sex-biased miRNAs in a healthy population. This study focused on disease-associated sex-biased miRNAs and shed light on the gender distinctions that manifest in pathological conditions. However, existing research has predominantly centered around mRNAs in pathological states, often overlooking the differential expression of miRNAs between genders in such conditions. Yet, these miRNAs could be potentially used in exploring novel medical technologies and therapies.

In this study, we identified 1264 FBmiRs and 431 MBmiRs, primarily examining samples associated with infections and neoplasms. Notably, we observed that FBmiRs displayed higher conservation with elevated DSW scores, indicative of divergent immune responses in infections and neoplasms compared to MBmiRs.

Nevertheless, it is essential to acknowledge that, while we collected numerous samples for this study, the number of samples per disease might still be insufficient, potentially introducing some bias. Additionally, the available data on miRNAs remain limited, which may affect the accuracy of certain findings. In addition, it should be noted that the expression of each dataset could vary enormously, which makes it difficult to compare different datasets. To avoid the addressed problem, in this study, we simply used the normalized miRNA expression profiles for each dataset. This procedure could present some bias. Additionally, it should be noted that we did not perform multiple testing correction in the identification of disease associated sex-biased miRNAs. The reason is that the aim of this study was not to find specific miRNA for further investigation but to perform a global and systematic analysis. Thus, we need to identify more miRNAs in the analysis. This procedure may also result in some bias.

## Figures and Tables

**Figure 1 genes-14-01688-f001:**
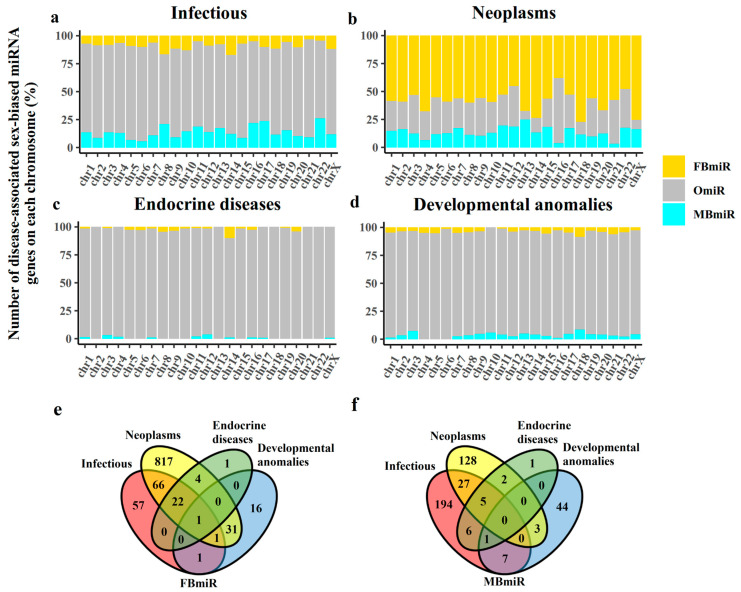
Number of disease-associated sex-biased miRNAs on each chromosome. This figure shows the distribution of FBmiR and MBmiR on each chromosome in the four disease groups. (**a**) Infectious, (**b**) Neoplasms, (**c**) Endocrine diseases, and (**d**) Developmental anomalies. (**e**) The overlap of FBmiR in each group. (**f**) The overlap of MBmiR in each group.

**Figure 2 genes-14-01688-f002:**
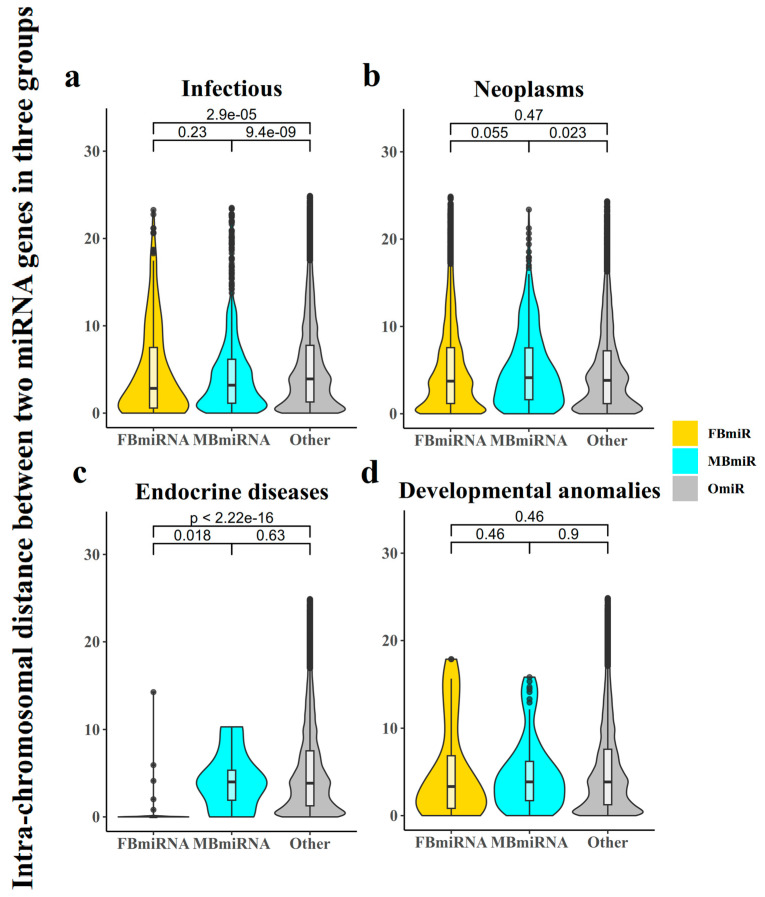
Intrachromosomal distance between any two miRNAs in three groups (FBmiR, MBmiR, and OmiR). Here, we calculated the distance between FBmiR and MBmiR for each pair on the same chromosome and performed Wilcoxon Rank Sum tests on the four disease groups: (**a**) Infectious, (**b**) Neoplasms, (**c**) Endocrine diseases, (**d**) Developmental anomalies.

**Figure 3 genes-14-01688-f003:**
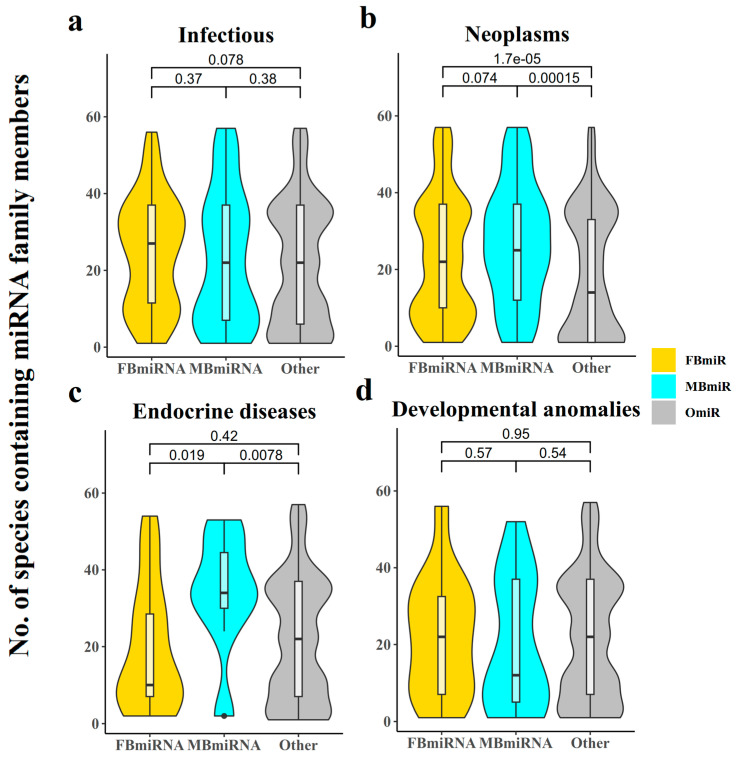
Number of species containing miRNA family members. This chart indicates how many confirmed species occur in each FBmiR or MBmiR in the four disease groups: (**a**) Infectious, (**b**) Neoplasms, (**c**) Endocrine diseases, (**d**) Developmental anomalies.

**Figure 4 genes-14-01688-f004:**
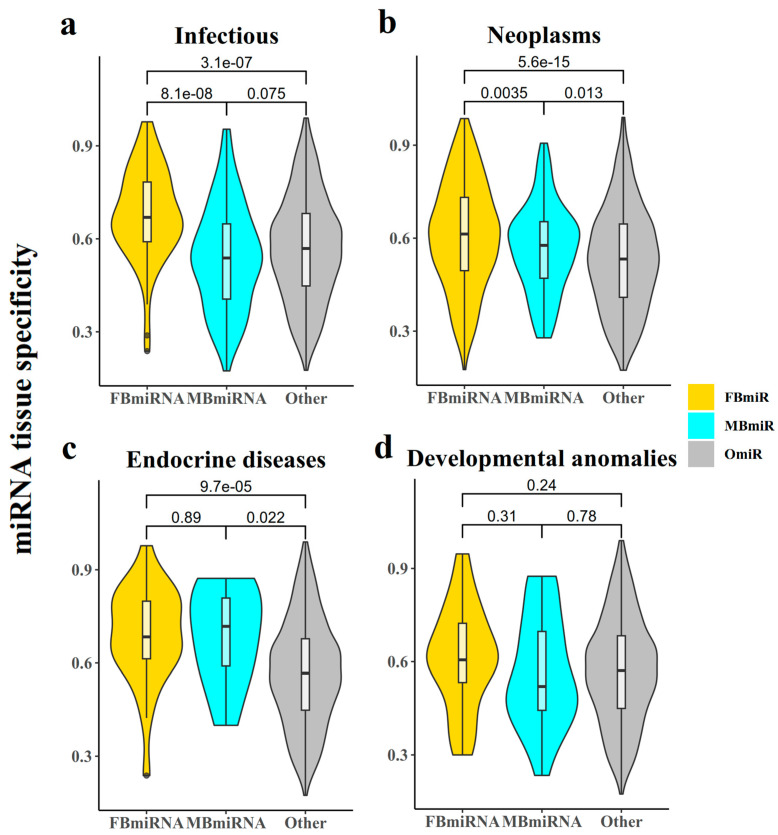
miRNA tissue specificity. This chart represents the tissue specificity score for each FBmiR or MBmiR in the four disease groups: (**a**) Infectious, (**b**) Neoplasms, (**c**) Endocrine diseases, and (**d**) Developmental anomalies.

**Figure 5 genes-14-01688-f005:**
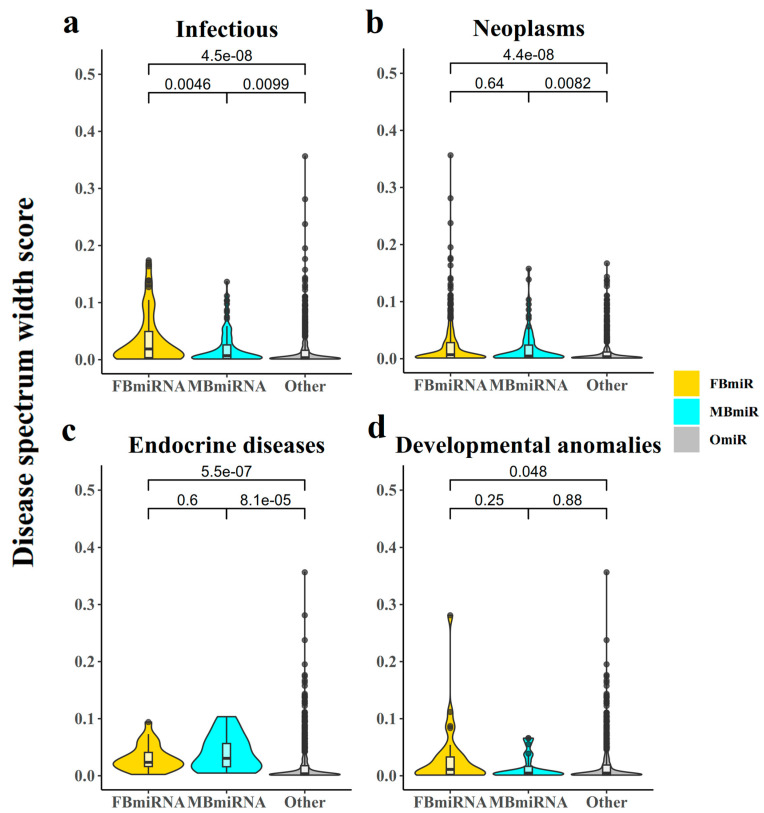
Disease spectrum width (DSW) score. This chart represents the disease spectrum width score for each FBmiR or MBmiR in the four disease groups: (**a**) Infectious, (**b**) Neoplasms, (**c**) Endocrine diseases, and (**d**) Developmental anomalies.

**Figure 6 genes-14-01688-f006:**
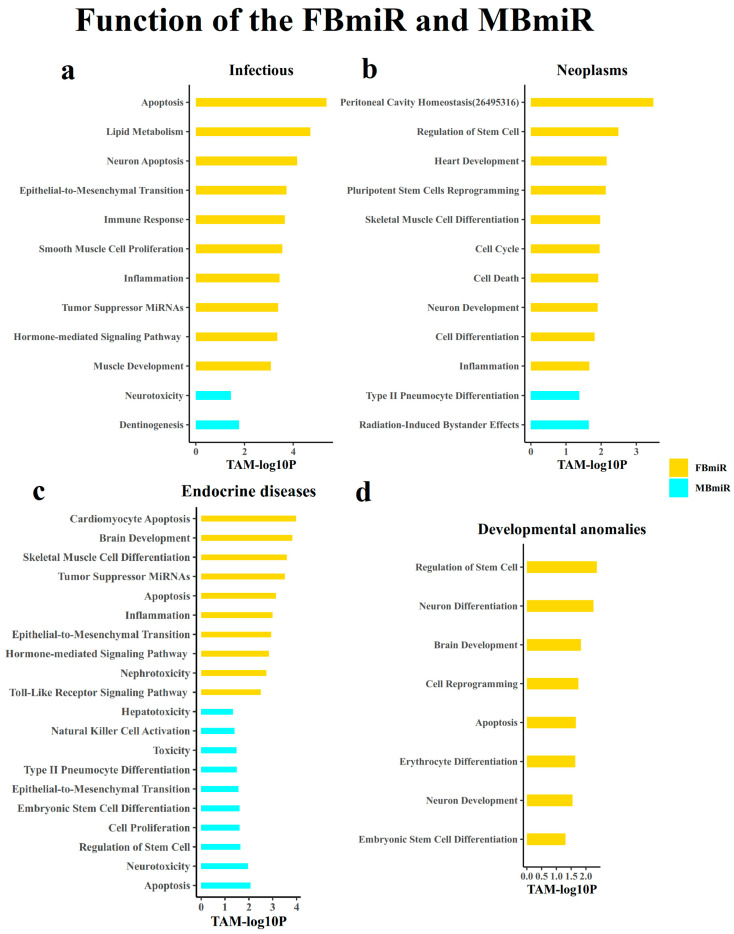
FBmiR- and MBmiR-related function enriched using the TAM2.0 algorithm in the four disease groups: (**a**) Infectious, (**b**) Neoplasms, (**c**) Endocrine diseases, and (**d**) Developmental anomalies.

**Table 1 genes-14-01688-t001:** Summary of the disease-associated sex-biased miRNAs and GEO source of the datasets used in this study.

Tissue	Disease	Dataset ID	Number of Samples		Number of miRNAs	
			Female	Male	FBmiR	MBmiR
Peripheral blood	pulmonary tuberculosis and sarcoidosis	GSE34608	28	54	26	5
	whole blood leukocytes	GSE134358	167	308	136	191
	lung cancer	GSE137140	749	997	1189	216
	cytogenetically normal acute myeloid leukemia	GSE142699	11	13	4	0
	Dementia	GSE167559	41	43	0	15
	transposition of the great arteries and systemic left and right ventricles	GSE215940	9	22	3	0
Brain	surgical specimens of primary glioblastoma multiform	GSE25631	28	54	26	5
	pediatric brain tumors	GSE42657	24	33	16	3
	glioma-related epilepsy	GSE199759	1	8	0	0
	temporal lobe epilepsy with hippocampal sclerosis	GSE205661	3	3	0	0
Neoplasms	human osteosarcoma (biopsies)	GSE39040	35	30	11	6
	human osteosarcoma	GSE39052	15	11	16	15
	locally advanced rectal cancer (LARC)	GSE68204	12	25	0	0
	HB tumors	GSE75283	31	27	9	4
	hepatocellular carcinoma with portal vein tumor thrombosis	GSE76903	9	51	88	17
	non-small cell lung cancer	GSE102286	38	52	5	0
	squamous cell carcinoma	GSE124678	9	23	7	0
	colorectal cancer	GSE126093	4	6	3	1
	Small Cell Bladder Cancer	GSE145259	3	19	0	0
	Stage I and II Clear Cell Renal Cell Carcinomas	GSE155209	2	18	0	0
Other	insulin resistance in primary human adipocytes	GSE41223	28	54	26	5
	congenital aniridia	GSE137995	15	5	55	25
	diabetic retinopathy	GSE160308	29	27	9	0
	healing and non-healing wounds	GSE174661	6	4	7	0

## Data Availability

Data are available at the download page of the HMDD database (http://www.cuilab.cn/hmdd (accessed on 29 March 2023)).

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
