# Peer review of "Identification and Analysis of Sex-Biased MicroRNAs in Human Diseases"

_genes, 2023, doi:10.3390/genes14091688_

Round 1

Reviewer 1 Report

First, I would like to acknowledge the authors' work and applaud their efforts in conducting this study. It is evident that the authors are highly recognized in the field, and this work seamlessly fits into their previous contributions. One particular strength of the manuscript is the authors' focus on investigating sex-biased microRNAs in diseases, as opposed to the prevalent approach of studying healthy cohorts or only one or at maximum few diseases. This approach is commendable, as it sheds light on potential sex-specific regulatory mechanisms underlying disease conditions.

To enhance the manuscript further, I would suggest that the authors compare their results in diseases to the findings in healthy cohorts. For instance, they could consider referencing and discussing articles like the ones with DOIs: https://doi.org/10.1038/s41598-018-35974-4 and https://academic.oup.com/clinchem/article/60/9/1200/5621677?login=true. This comparison could offer valuable insights into the distinct miRNA profiles between healthy and diseased individuals.

Additionally, the specific miRNAs identified in the study could be discussed in greater detail. Presenting highlights and discussing the biological significance of these miRNAs would provide a more comprehensive understanding of their potential roles in disease. The TAM analysis is a great step in this direction but maybe it would be worth having it also on the single miRNA level.

Furthermore, it would be beneficial if the authors could address potential biases in their study. They might consider adding information about the technologies used for the miRNA expression analysis (e.g., RT-qPCR data, microarray data, and NGS data) in Table 1. Acknowledging and discussing these technologies can help readers interpret the results more accurately and assess their impact on the findings.

Overall, this manuscript makes a valuable contribution to the field of sex-specific miRNA expression in human diseases. With the suggested revisions and additions, it has the potential to become an even more informative and impactful study. The insights gained from this work could have significant implications for precision medicine and the development of targeted approaches to disease management.

Author Response

Reviewer 1

General Comment: First, I would like to acknowledge the authors' work and applaud their efforts in conducting this study. It is evident that the authors are highly recognized in the field, and this work seamlessly fits into their previous contributions. One particular strength of the manuscript is the authors' focus on investigating sex-biased microRNAs in diseases, as opposed to the prevalent approach of studying healthy cohorts or only one or at maximum few diseases. This approach is commendable, as it sheds light on potential sex-specific regulatory mechanisms underlying disease conditions.

Response: We thank the reviewer for this comment.

Comment: To enhance the manuscript further, I would suggest that the authors compare their results in diseases to the findings in healthy cohorts. For instance, they could consider referencing and discussing articles like the ones with DOIs: https://doi.org/10.1038/s41598-018-35974-4 and https://academic.oup.com/clinchem/article/60/9/1200/5621677?login=true. This comparison could offer valuable insights into the distinct miRNA profiles between healthy and diseased individuals.

Response: It is a good suggestion. As suggested, we performed analysis and provided the results in the revised manuscript.

Comment: Additionally, the specific miRNAs identified in the study could be discussed in greater detail. Presenting highlights and discussing the biological significance of these miRNAs would provide a more comprehensive understanding of their potential roles in disease. The TAM analysis is a great step in this direction but maybe it would be worth having it also on the single miRNA level.

Response: It is a good suggestion. As suggested, we provided detailed discussion for some specific miRNAs in the revised manuscript.

Comment: Furthermore, it would be beneficial if the authors could address potential biases in their study. They might consider adding information about the technologies used for the miRNA expression analysis (e.g., RT-qPCR data, microarray data, and NGS data) in Table 1. Acknowledging and discussing these technologies can help readers interpret the results more accurately and assess their impact on the findings.

Response: Yes, potential bias could be existed in omics data. We discussed this in the revised manuscript. We thank the reviewer for this point.

Comment: Overall, this manuscript makes a valuable contribution to the field of sex-specific miRNA expression in human diseases. With the suggested revisions and additions, it has the potential to become an even more informative and impactful study. The insights gained from this work could have significant implications for precision medicine and the development of targeted approaches to disease management.

Response: We thank the reviewer for the comment.

Reviewer 2 Report

The paper is dealing with miRNAs and their loci, in several species, for its sex-bias in health or disease. The topic is of immense importance.

The authors mention that they dismissed folding changes. I wonder about endogenous controls.

I am not sure if it is scientific possible to compare Expression levels without mention the endogenous controls. For me its not clear if all data sets used the same endogenous controls. If not, the expression of each set would vary enormous. And couldn’t be compared.

How raw, was the raw data? Were all miRNAs calculated using the same endogenous control/s?
To my knowledge this is important to compare the expression levels, because there is no endogenous miRNA which plays no functional role.

Was ∆Ct used for calculation? Was a global expression used as endogenous control?

Sex-Specific miRNA Differences in Liquid Biopsies from Subjects with Solid Tumors and Healthy Controls could be added as relevant information in the introduction.

The texts in the figures are too small. Especially fig 1 a and b and fig 6. But also the captions in all figures.  

If the endogenous controls were considered, the paper is in a good shape and of good understanding. Only the methods would need to be more detailed.

If the endogenous controls were not considered, the results would be somehow random, and everything would have to be recalculated.

Author Response

Comment: The paper is dealing with miRNAs and their loci, in several species, for its sex-bias in health or disease. The topic is of immense importance.

Response: We thank the reviewer for this comment.

Comment: I am not sure if it is scientific possible to compare Expression levels without mention the endogenous controls. For me its not clear if all data sets used the same endogenous controls. If not, the expression of each set would vary enormous. And couldn’t be compared.

Response: We thank the reviewer for this comment. We agree with the reviewer’s opinion. Different datasets often resulted from different platforms, different laboratories, different technicians etc, which could produce significantly different expression level and batch effects. So, in this study, we did not compare between or among different datasets. We only identified disease associated sex-biased miRNAs in each dataset. We discussed the proposed concern in the revised manuscript.

Comment: How raw, was the raw data? Were all miRNAs calculated using the same endogenous control/s?

To my knowledge this is important to compare the expression levels, because there is no endogenous miRNA which plays no functional role.

Was ∆Ct used for calculation? Was a global expression used as endogenous control?

Response: We thank the reviewer for this comment. We agree with the reviewer’s opinion. Yes, the topic is quite important. Different datasets often resulted from different platforms, different laboratories, different technicians etc, which could produce significantly different expression level and batch effects. We did not use ∆Ct and use specific endogenous control. To avoid the addressed problem, in this study we just used the normalized miRNA expression profiles for each dataset.

Comment: Sex-Specific miRNA Differences in Liquid Biopsies from Subjects with Solid Tumors and Healthy Controls could be added as relevant information in the introduction.

Response: It is a good suggestion. As suggested, we added the related information in the Introduction of the revised manuscript.

Comment: The texts in the figures are too small. Especially fig 1 a and b and fig 6. But also the captions in all figures. 

Response: We thank the reviewer for the suggestion. We modified the figures to make the texts clearer.

Comment: If the endogenous controls were considered, the paper is in a good shape and of good understanding. Only the methods would need to be more detailed.

If the endogenous controls were not considered, the results would be somehow random, and everything would have to be recalculated.

Response: We thank the reviewer for this comment. We agree that endogenous controls have significant effects on the analysis of the miRNA expression data. To avoid the addressed problem, in this study we just used the normalized miRNA expression profiles in each specific dataset. Different datasets may use different protocol for data normalization. 

Reviewer 3 Report

In this manuscripot, Cui et al reanalyze 24 disease miRNA datasets from the GEO that include male and female samples to identify miRNAs that are differentially expressed between males and females (difMIRs). The authors identify their distribution across chromosomes, report the overlap of difMIRs found for different diseases, report the conservation of difMIRs across species, and carry out functional and cell type enrichment.

Major comments:

1)     A major novelty claim of the present work is that sex-specific miRNA comparisons have largely been done in healthy individuals and not disease states. However, multiple articles not cited have carried out this analysis, for example (1-4) out of countless examples.

2)     The analyses are presented more as a disjointed list of items, without context, justification, or insight. The manuscript lacks biological message or interpretation.

3)     Analyses are presented without multiple testing correction, including genome-wide mirDIF and the enrichment analyses, and most of the results are likely not statistically significant.

1.     Guo, L., Zhang, Q., Ma, X. et al. miRNA and mRNA expression analysis reveals potential sex-biased miRNA expression. Sci Rep 7, 39812 (2017). 

2.     Florijn, B.W., Valstar, G.B., Duijs, J.M.G.J. et al. Sex-specific microRNAs in women with diabetes and left ventricular diastolic dysfunction or HFpEF associate with microvascular injury. Sci Rep 10, 13945 (2020)

3.     Florijn, B.W.; Bijkerk, R.; Kruyt, N.D.; van Zonneveld, A.J.; Wermer, M.J.H. Sex-Specific MicroRNAs in Neurovascular Units in Ischemic Stroke. Int. J. Mol. Sci. 2021, 22, 11888

4.     Queirós, A. M. et al. Sex- and estrogen-dependent regulation of a miRNA network in the healthy and hypertrophied heart. International Journal of Cardiology 169, 331–338 (2013).

The manuscript needs extensive editing for language, as in many cases it is incoherent, preferably by someone who is an expert in the field. In many cases the language is not professional enough, and in other cases, terms are used wrongly (e.g: "folding changes" and others). 

Author Response

Comment: In this manuscripot, Cui et al reanalyze 24 disease miRNA datasets from the GEO that include male and female samples to identify miRNAs that are differentially expressed between males and females (difMIRs). The authors identify their distribution across chromosomes, report the overlap of difMIRs found for different diseases, report the conservation of difMIRs across species, and carry out functional and cell type enrichment.

Response: We thank the reviewer for this comment.

Comment: A major novelty claim of the present work is that sex-specific miRNA comparisons have largely been done in healthy individuals and not disease states. However, multiple articles not cited have carried out this analysis, for example (1-4) out of countless examples.

  1. Guo, L., Zhang, Q., Ma, X. et al. miRNA and mRNA expression analysis reveals potential sex-biased miRNA expression. Sci Rep 7, 39812 (2017).

  1. Florijn, B.W., Valstar, G.B., Duijs, J.M.G.J. et al. Sex-specific microRNAs in women with diabetes and left ventricular diastolic dysfunction or HFpEF associate with microvascular injury. Sci Rep 10, 13945 (2020)

  1. Florijn, B.W.; Bijkerk, R.; Kruyt, N.D.; van Zonneveld, A.J.; Wermer, M.J.H. Sex-Specific MicroRNAs in Neurovascular Units in Ischemic Stroke. Int. J. Mol. Sci. 2021, 22, 11888

  1. Queirós, A. M. et al. Sex- and estrogen-dependent regulation of a miRNA network in the healthy and hypertrophied heart. International Journal of Cardiology 169, 331–338 (2013).

Response: We thank the reviewer for this comment and sorry for not comprehensively found the needed references. In the revised manuscript, we added these references.

Comment: The analyses are presented more as a disjointed list of items, without context, justification, or insight. The manuscript lacks biological message or interpretation.

Response: We thank the reviewer for this comment. We added more descriptions and interpretation for the analysis and results.

Comment: Analyses are presented without multiple testing correction, including genome-wide mirDIF and the enrichment analyses, and most of the results are likely not statistically significant.

Response: We thank the reviewer for this comment and agree with the reviewer. Yes, we knew this problem when performing the analysis. The reason why we did not perform multiple testing correction is that the aim of this study is not to find specific miRNA for further investigation but to perform a global and systematic analysis. In order to identify more miRNAs, we did not use multiple testing correction. We know this could increase the false positives for specific miRNAs but would decrease false negatives as well. We know it is a tradeoff in the accuracy of specific miRNAs and systematic analysis, which needs a number of miRNAs. We discussed this in the revised manuscript. Thanks again for this comment.

Comment: The manuscript needs extensive editing for language, as in many cases it is incoherent, preferably by someone who is an expert in the field. In many cases the language is not professional enough, and in other cases, terms are used wrongly (e.g: "folding changes" and others).

Response: We thank the reviewer for this comment. As suggested, we asked one native English speaker (english-69820 by MDPI) to edit the manuscript.

Round 2

Reviewer 1 Report

thank you for addressing all comments